# Parameter-Efficient Generation of Natural Language Explanations for Chest X-ray Classification

**Isabel Rio-Torto**[1,3]                                   ISABEL.RIOTORTO@INESCTEC.PT

**Jaime S. Cardoso**[2,3]                                            JSC@FE.UP.PT

**Luís F. Teixeira**[2,3]                                           LUISFT@FE.UP.PT

[1]*Departamento de Ciência de Computadores, Faculdade de Ciências, Universidade do Porto, Rua do Campo Alegre s/n, 4169–007 Porto, Portugal*

[2]*Faculdade de Engenharia, Universidade do Porto, Rua Dr. Roberto Frias s/n, 4200-465 Porto, Portugal*

[3]*INESC TEC, Campus da FEUP, Rua Dr. Roberto Frias s/n, 4200-465 Porto, Portugal*

**Editors:** Accepted for publication at MIDL 2024

## Abstract

The increased interest and importance of explaining neural networks' predictions, especially in the medical community, associated with the known unreliability of saliency maps, the most common explainability method, has sparked research into other types of explanations. Natural Language Explanations (NLEs) emerge as an alternative, with the advantage of being inherently understandable by humans and the standard way that radiologists explain their diagnoses. We extend upon previous work on NLE generation for multi-label chest X-ray diagnosis by replacing the traditional decoder-only NLE generator with an encoder-decoder architecture. This constitutes a first step towards Reinforcement Learning-free adversarial generation of NLEs when no (or few) ground-truth NLEs are available for training, since the generation is done in the continuous encoder latent space, instead of in the discrete decoder output space. However, in the current scenario, large amounts of annotated examples are still required, which are especially costly to obtain in the medical domain, given that they need to be provided by clinicians. Thus, we explore how the recent developments in Parameter-Efficient Fine-Tuning (PEFT) can be leveraged for this use-case. We compare different PEFT methods and find that integrating the visual information into the NLE generator layers instead of only at the input achieves the best results, even outperforming the fully fine-tuned encoder-decoder-based model, while only training 12% of the model parameters. Additionally, we empirically demonstrate the viability of supervising the NLE generation process on the encoder latent space, thus laying the foundation for RL-free adversarial training in low ground-truth NLE availability regimes. The code is publicly available at https://github.com/icrto/peft-nles.

**Keywords:** parameter-efficient, natural language explanations, chest x-ray diagnosis

## 1. Introduction

The need for providing explanations is essential in domains like medicine, given the life-or-death nature of the decisions involved (Vayena et al., 2018). Given their training-free nature, saliency maps are, by far, the most commonly employed explanation method, despite their known problems (Adebayo et al., 2018; Camburu et al., 2019; Rudin, 2019).

Among other explanation modalities are Natural Language Explanations (NLEs). These are textual descriptions of the decision-making process of a neural network for a given input

image, thus combining image descriptions (e.g. obtained via image captioning models) - *image-relevance* - and model predictions - *class-relevance* (Hendricks et al., 2016; Rio-Torto et al., 2022). NLEs are intrinsically understandable by humans and can provide complementary insights to those of other types of explanations. For example, while visual explanations are more spatially precise, NLEs can be more semantically meaningful (Rio-Torto et al., 2022). In the medical domain in general, and in chest X-ray diagnosis in particular, NLEs can be even more useful, given that they correspond to how radiologists explain their diagnoses (Gale et al., 2018; Miller, 2019).

There are several works on NLEs for natural images (Hendricks et al., 2016; Park et al., 2018; Wu and Mooney, 2019; Do et al., 2020; Marasović et al., 2020; Kayser et al., 2021; Majumder et al., 2022; Plüster et al., 2022; Sammani et al., 2022) and for text-only models (Camburu et al., 2018; Rajani et al., 2019; Kumar and Talukdar, 2020; Narang et al., 2020); they usually apply the traditional cross entropy loss on the generated NLEs and condition the NLE generator on information from the predictions of the model that is being explained. We highlight the work by Hendricks et al. (Hendricks et al., 2016) since it explicitly imposes class-relevance on the generated NLEs through the proposal of a Reinforcement Learning-based (RL) loss. Inspired by this and by the work on adversarial text generation (Donahue and Rumshisky, 2018), **we propose replacing the traditional decoder-only NLE generator with an encoder-decoder architecture** (see Figure 1). This paves the way for the RL-free adversarial generation of NLEs, when no (or few) ground-truth data is available to train the model.

In the medical domain, text generation mostly consists of report generation approaches (Hou et al., 2021; Monshi et al., 2020; Ouis and A. Akhloufi, 2024). To the best of our knowledge, Kayser et al. (Kayser et al., 2022) is the first and only work on NLE generation for the medical domain. Therefore, we build upon this work; namely, we use the dataset the authors introduce, MIMIC-NLE, and their proposed evaluation framework.

However, contrary to the models proposed by Kayser et al. (Kayser et al., 2022), **we adopt Parameter-Efficient Fine-Tuning (PEFT) when training our NLE generator** to receive visual features and classification predictions. Instead of fine-tuning all model parameters, PEFT methods fine-tune only a small percentage, while still performing competitively (Zhang et al., 2023). A plethora of PEFT methods has been proposed, not only for single modality use-cases (Rebuffi et al., 2017; Houlsby et al., 2019; Lester et al., 2021; Li and Liang, 2021; Liu et al., 2021; Chen et al., 2023; Yu et al., 2023; Zhang et al., 2023), but also for multi-modal applications (Luo et al., 2023; Zhang et al., 2023; Gao et al., 2024), and in the medical domain as well (Dutt et al., 2023). Applying PEFT to NLE generation is still an emergent topic, with work only on uni-modal applications (Solano et al., 2023).

Therefore, our contributions are the following:

- Replacement of the decoder-only NLE generator of Kayser et al. (Kayser et al., 2022) by an encoder-decoder architecture that lays the foundation for the RL-free adversarial generation of NLEs when no (or few) ground-truth data is available during training.

- Introduction of the first work exploring different PEFT methods for NLE generation for chest X-ray classification.

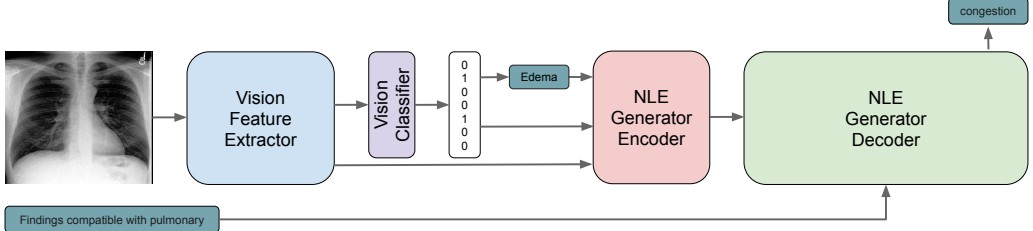

Figure 1: Block diagram of our NLE generation architecture for the pathology "Edema". The encoder receives the image features and the prediction information, while the decoder generates the NLE from the latent space of the encoder.

## 2. Methodology

### 2.1. MIMIC-NLE Dataset

NLEs are fundamentally different from image descriptions, since the former need to be not only image-relevant but also decision-relevant, i.e. take into account the decision of the underlying model they are trying to explain (Hendricks et al., 2016; Rio-Torto et al., 2022). Although there are some datasets for chest X-ray diagnosis that include medical reports in text form (i.e. image descriptions) (Demner-Fushman et al., 2015; Johnson et al., 2019; Bustos et al., 2020), the work by Kayser et al. (Kayser et al., 2022) was the first and, to the best of our knowledge, the only one to propose a dataset with NLEs.

The MIMIC-NLE dataset (Kayser et al., 2022) is obtained automatically from the MIMIC-CXR dataset (Johnson et al., 2019) by using a BERT-based labeler in conjunction with clinically validated extraction rules. The dataset comprises 38003 image-NLE pairs or 44935 image-diagnosis-NLE triplets (some NLEs explain multiple diagnoses). Note that each image can simultaneously be associated with 10 pathologies (*Atelectasis*, *Consolidation*, *Edema*, *Enlarged Cardiomediastinum*, *Lung Lesion*, *Lung Opacity*, *Pleural Effusion*, *Pleural Other*, *Pneumonia*, and *Pneumothorax*). The data is split into 37016, 273, and 714 NLEs for training, validation and testing, respectively.

### 2.2. Baselines

Besides introducing MIMIC-NLE, Kayser et al. (Kayser et al., 2022) also tested different architectures to generate NLEs for this multi-label scenario. These architectures, which we describe below, are used here as baselines to compare our proposals with. All architectures work in a similar fashion: a vision model (DenseNet-121 (Huang et al., 2017)) extracts features and classifies the input image; the features and a vector representation of the prediction, together with the pathology for which the NLE is being generated, are fed into an autoregressive language model, which generates the NLE. During training, instead of using the prediction vector, the ground-truth label is used, and the vision model is trainable. Thus, this can be considered an in-model explanation architecture that works within a predict-explain paradigm (Do et al., 2020).

RAdiological Text Captioning for Human Examined Thoraxes (RATCHET) was first proposed by Hou et al. (Hou et al., 2021) as a chest X-ray captioning method. Kayser et al. (Kayser et al., 2022) leverage this architecture to generate NLEs instead. It consists of

a GPT-2 (Radford et al., 2018) model with cross-attention layers, that receive the vision classifier's features concatenated with a projection of the prediction vector, as depicted in Figure 2 (top left). As input, the model receives the NLE-related pathology.

DPT (DenseNet-121 + GPT-2) is proposed by Kayser et al. (Kayser et al., 2022) and it consists of a GPT-2 model that receives as input the vision features, the prediction vector, and the NLE-related pathology. An illustration of this architecture can be found in Figure 2 (top right).

### 2.3. Proposed Approach

**Architecture** Our approach builds upon the work of Kayser et al. (Kayser et al., 2022), but replaces the NLE decoder with a denoising auto-encoder (see Figure 1). Not only does this increase the representation capabilities of the NLE generator, but, most importantly, it lays the foundation for the RL-free adversarial generation of NLEs when no (or few) ground-truth NLEs are available for training. In this scenario, teacher forcing is unfeasible. While an adversarial approach can solve the problem, if the generator is a decoder-only architecture (e.g. DPT), RL is needed, since its output (and the input to the discriminator) is discrete and, as such, no gradient would flow from the discriminator to the generator (Donahue and Rumshisky, 2018). By using an encoder-decoder architecture, where the adversarial learning happens on the continuous latent space of the encoder at the sentence level, gradient propagation is ensured without needing RL, thus making the training process faster (obtaining an NLE now only involves a single encoder forward pass instead of N decoder passes, where N is the number of tokens in the generated sentence) and more stable. Finally, decoding this latent space into text to obtain the NLE is still possible thanks to the pre-trained decoder.

We use the auto-encoder proposed by (Montero et al., 2021), which consists of a frozen BERT-based encoder, a multi-head attention mechanism as a bottleneck layer, and a single transformer decoder layer with cross attention. Since this cross attention layer receives the same input at every timestep, the authors introduce a gating mechanism that controls the amount of information to keep at each timestep. Trained on MIMIC-NLE (Kayser et al., 2022), it achieves a reconstruction BLEU-4 score of 93.9.

**PEFT** Our auto-encoder is trained on text inputs, but it needs to be adapted to incorporate the products of the vision model we want to explain. Recently, the transfer learning paradigm has shifted from fine-tuning entire models to only certain layers. This is especially important in NLE generation, and even more so in post-model methods, where given a pre-trained model, we want to obtain explanations for its decision process with the shortest training time possible. Therefore, we experiment with applying several PEFT techniques to the encoder of our NLE Generator, while keeping its decoder module frozen. These methods are depicted in Figure 2 (last three rows).

In total, we try five PEFT methods, four of which are built on top of an input configuration similar to that of DPT, i.e. where both image features, prediction vector and associated pathology are given as inputs to the encoder.

LoRA (Yu et al., 2023) represents, through low-rank decomposition, the model's weight updates into two smaller matrices, which are trained, while the original model weights are kept frozen. After training, both weight matrices are combined.

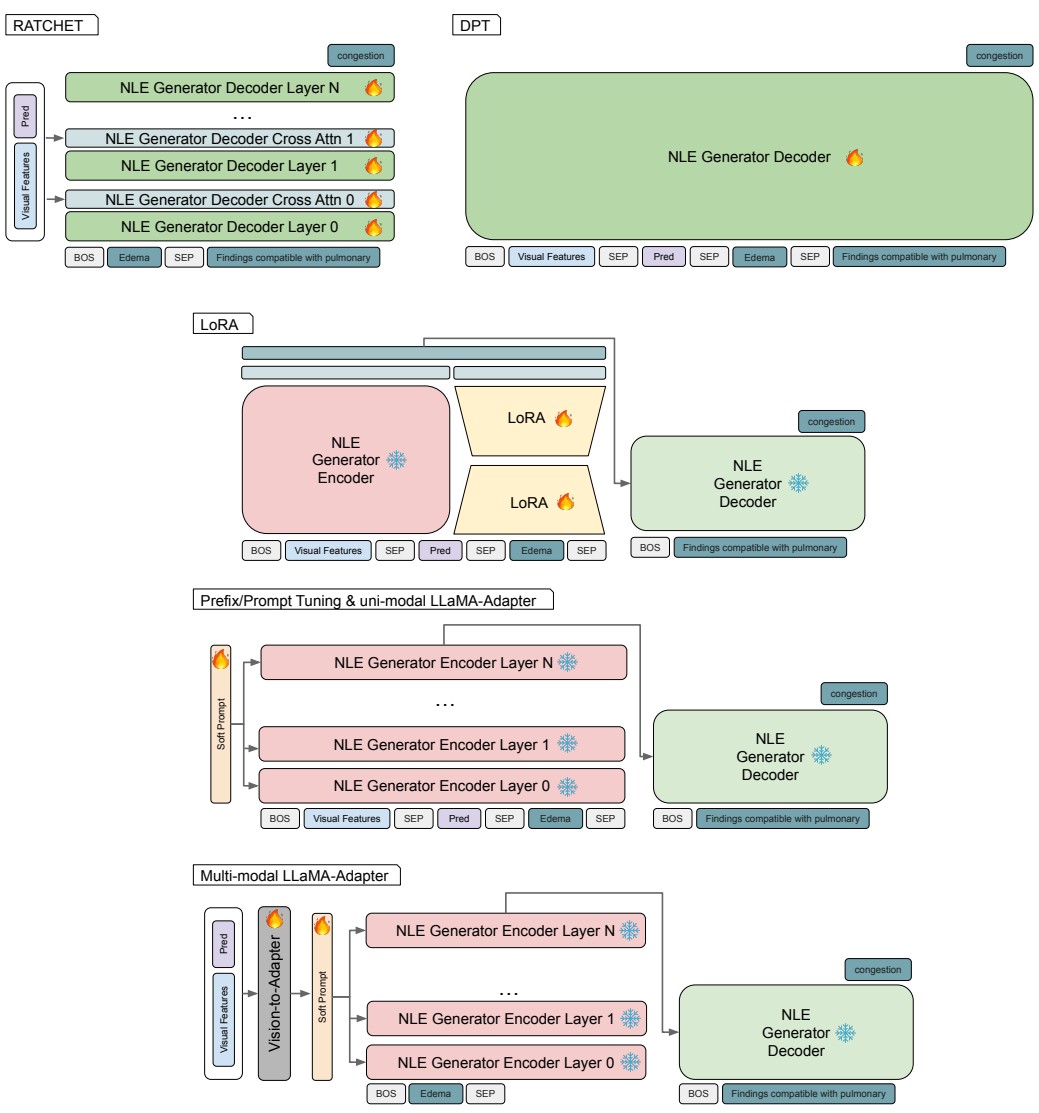

Figure 2: The different types of (PEFT) architectures of the NLE generator. Trainable layers/parameters are represented by the fire icon, while frozen blocks are represented by the snowflake.

Prompt and Prefix Tuning (Lester et al., 2021; Li and Liang, 2021) insert a given number of learnable soft prompts into the frozen model. While Prompt Tuning (Lester et al., 2021) only appends these prompts to the input embeddings, Prefix Tuning (Li and Liang, 2021) applies these prompts to several layers of the frozen model (in our case, to all layers).

LlaMA-Adapter (Zhang et al., 2023) was proposed to efficiently fine-tune a large language model, LlaMA (Touvron et al., 2023), and it differs from Prefix Tuning in the way the learnable adapters are merged into the frozen model. The authors propose to separate the adapter prompts from the word tokens and introduce a zero-initialized gating mechanism that adaptively controls the influence of the adapter prompts during training. This way,

in the early training stages, the influence of the adapter prompt is very small, and it is progressively increased, which guarantees training stability while still ensuring the model is gradually being adapted and simultaneously retaining the knowledge of the original model.

However, none of the aforementioned methods take into account the multi-modality of our scenario, as they were developed for uni-modal applications. To tackle this, the authors of LlaMA-Adapter also propose a multi-modal version. Instead of providing the image-related information through the input of the model, it is directly added to the adaptation tokens. To ensure the image features have the same dimensions as the adaptation prompts, a Transformer encoder is used, as shown in Figure 2 (last row).

**NLE Generation Loss Function**  As previously mentioned, using an encoder-decoder architecture opens the door to the RL-free adversarial generation of NLEs in low ground-truth availability regimes. To test the viability of such an approach despite having, as we do in this work, ground-truth NLEs for the whole dataset, we experiment with supervising the NLE generation directly in the latent space of the encoder, as this is where the adversarial learning would happen. Thus, in this experiment, we replace the cross entropy loss computed between the discrete output of the decoder and the ground-truth NLE, by the Mean Squared Error (MSE) computed between the encoder latent representation of the ground-truth NLE and the continuous output of the encoder.

## 3. Experimental Setup

**Evaluation Metrics**  We follow the evaluation protocol proposed in the MIMIC-NLE paper (Kayser et al., 2022); several aspects are evaluated, namely the chest X-ray multi-label classification performance (through balanced AUC - Area Under the Curve), and the NLEs' quality (only for correctly predicted instances). Given that there is evidence that automated natural language generation (NLG) metrics correlate badly with the way humans evaluate how similar two NLEs are (Kayser et al., 2021), because two NLEs can be simultaneously correct while having very different syntactic forms, the authors propose the CLEV (CLinical EVidence) score. This score uses the CheXbert labeler (Smit et al., 2020) to verify if an NLE refers to the same clinical evidence as the ground-truth NLE. Thus, it corresponds to the accuracy regarding clinical evidence. Several NLG metrics are considered, with a bigger importance given to BERTScore (Zhang et al., 2020) and METEOR (Lavie and Agarwal, 2007) since it has been shown that these exhibit the highest correlation with human evaluation (Kayser et al., 2021).

**Implementation Details**  Regarding the pre-training of the auto-encoder, the decoder is a single transformer layer, as proposed in the original AUTOBOT paper (Montero et al., 2021), and the encoder is the CXR-BERT model (Boecking et al., 2022). CXR-BERT is a BERT masked language model trained on PubMed abstracts (U.S. National Library of Medicine), clinical notes from MIMIC-III (Johnson et al., 2016) and MIMIC-CXR (Johnson et al., 2019). The model is trained with the AdamW optimizer for 1 million steps with a batch size of 64 and a linearly decayed learning rate of 0.001 with 1000 warmup steps. The best model is chosen based on the reconstruction BLEU-4.

For the explanation generation model, we follow the training details described in the MIMIC-NLE paper (Kayser et al., 2022): for RATCHET, the learning rate is $5 \times 10^{-5}$, and

$5 \times 10^{-4}$ for all remaining experiments. All models are trained with the AdamW optimizer for 50 epochs with a batch size of 16 and a linearly decayed learning rate with 1000 warmup steps. We use Early Stopping with a patience of 25.

LoRA parameters follow the original LoRA paper (Yu et al., 2023), i.e. an alpha of 32 and a dropout of 0.01. For the remaining PEFT methods, the learnable prompts have 10 tokens (as in the original LlaMA-Adapter (Zhang et al., 2023)), and for both LlaMA-Adapter models, the prompts are injected at every layer of the NLE generator encoder. The transformer used in the multi-modal LlaMA-Adapter has the following configuration: 2 transformer layers with 4 attention heads, an embedding dimension of 768, and the linear layers have 3072 neurons.

The best model and corresponding hyperparameters are chosen with the same criteria as in the MIMIC-NLE paper (Kayser et al., 2022): the product of the classification score (balanced AUC), CLEV score, and the average of BERTSCore (Zhang et al., 2020) and METEOR (Lavie and Agarwal, 2007), computed on the validation set.

We learn the weighting scheme of the classification and text losses, following the methodology proposed in (Cipolla et al., 2018). Thus, our loss function is given by $\mathcal{L} = \mathcal{L}_{clf} + \frac{1}{\sigma^2}\mathcal{L}_{text} + log(\sigma)$, where $\sigma$ is learned during training and has a starting value of 1.0.

## 4. Results and Discussion

Table 1 contains the results of the different NLE generation frameworks evaluated in terms of balanced AUC, CLEV and NLG metrics. We also present the number of correctly classified instances (for which the CLEV and NLG metrics are computed), the number of additional trainable parameters introduced, and the overall performance.

Besides the base DenseNet-121 model, we present the results of two of the models of (Kayser et al., 2022), RATCHET and DPT, trained with our loss weighting scheme. The obtained results follow the trends verified in the original paper, i.e. RATCHET improves AUC over the DenseNet-121, while DPT does not, and the former is better overall than the latter.

Regarding our proposals, replacing the decoder-only NLE generator (i.e. DPT) with the encoder-decoder (Base AE) improves the overall performance: while the AUC slightly decreases, the CLEV and BERT scores increase. However, this comes at a cost: 145M trainable parameters are added, 20M more than DPT introduces.

Uni-modal PEFT techniques slightly decrease or perform on par with the Base AE model in terms of overall performance. However, these methods introduce much less trainable parameters (11M in the worst case). Given the small loss in performance but the high gain in training efficiency, PEFT methods are, in fact, suitable for NLE generation for chest X-ray diagnosis, as had already been shown in other domains. Our results also validate the findings of the original LLaMA-Adapter (Zhang et al., 2023), i.e. that the zero-initialized gating mechanism surpasses traditional Prefix Tuning.

The multi-modal version of LLaMA-Adapter achieves the best overall results among our proposals, even surpassing the fully fine-tuned model. This confirms our hypothesis that incorporating the visual information throughout the NLE generator layers instead of just at the input is beneficial. Indeed, this is probably the same reason why RATCHET outperforms DPT. This result demonstrates that explainability and task performance are

Table 1: Results obtained on the MIMIC-NLE (Kayser et al., 2022) test set. Evaluation metrics include the balanced AUC, the CLEV score, and several NLG metrics. CLEV and NLG metrics are only computed for correctly classified instances. This value is presented in column # instances, for which the total is 835 (one NLE can explain more than one prediction). The # params column corresponds to the number of trainable parameters (in millions, M) relative to the 7M DenseNet-121 model. The last column shows the overall performance, given by the product of the AUC, the CLEV score, and the average of the BERT and METEOR scores. In bold are the best results for each metric across all models, while underlined are the best results among our models.

| Model | AUC | CLEV | BERTS | METEOR | BLEU-1 | BLEU-4 | ROUGE | CIDEr | SPICE | # instances | # params. (↓) | overall |
|---|---|---|---|---|---|---|---|---|---|---|---|---|
| DenseNet-121 | 64.45 | | | | | | | | | | 7M | |
| RATCHET | **65.19** | 74.81 | 77.49 | 13.28 | 19.92 | 3.933 | 21.53 | 37.71 | 20.04 | 540 | +90.5M | **221337** |
| DPT | 63.24 | 73.90 | 77.97 | 13.57 | 21.02 | **4.364** | **22.65** | **40.67** | 19.25 | 498 | +125M | 213903 |
| Base AE | 63.01 | 74.19 | **78.42** | 13.10 | 15.31 | 2.497 | 16.58 | 11.81 | 15.02 | 527 | +145M | 213915 |
| LoRA AE | 63.98 | 73.99 | 77.51 | **15.32** | 20.38 | 4.002 | 20.60 | 24.44 | **20.11** | 546 | +1.1M | 219723 |
| Prompt AE | 61.31 | 74.49 | 77.16 | 14.43 | 19.49 | 3.773 | 19.76 | 23.48 | 18.10 | 541 | **+0.8M** | 209145 |
| Prefix AE | 65.19 | 67.96 | 77.56 | 14.70 | 20.06 | 3.739 | 19.77 | 21.53 | 18.84 | **568** | +1.2M | 204370 |
| LLaMA-Adapt AE | 63.87 | 71.05 | 77.17 | 14.61 | **23.06** | 4.356 | 21.47 | 29.74 | 19.00 | 532 | +1.0M | 208247 |
| + multi-modal | 64.86 | 74.63 | 76.95 | 14.12 | 18.77 | 3.098 | 18.65 | 18.40 | 16.29 | 536 | +11M | 220412 |
| + MSE loss | 62.32 | 77.78 | 78.05 | 10.04 | 16.46 | 2.075 | 14.20 | 13.97 | 9.710 | 495 | +11M | 213497 |

not always at odds with each other and, in fact, explainability might improve classification accuracy.

Finally, we train the best PEFT configuration (multi-modal LLaMA-Adapter) directly with MSE on the latent space of the encoder. While the decrease in NLG metrics is expected since supervision now occurs at the sentence level instead of at the token level, the CLEV score significantly increases, which shows that, although the generated NLEs might by syntactically different from the ground-truth NLEs, they contain the correct clinical evidence. Thus, this confirms that our proposed approach is viable, laying the foundation for the RL-free adversarial generation of NLEs.

We present qualitative examples in Figure 3 of the Appendix. They corroborate the quantitative results, mainly that the PEFT approaches produce NLEs consistent with fully fine-tuned models, even while training only a fraction of the parameters, and that reasonable NLG metric values do not always correlate with correct clinical evidence.

## 5. Conclusion

We build upon the seminal work of Kayser et al. (Kayser et al., 2022), which tackled the problem of generating NLEs for chest X-ray diagnosis, by replacing the decoder-only NLE generator by an encoder-decoder, coupled with PEFT techniques to drastically reduce the number of trainable parameters while maintaining competitive performance. We show that using the multi-modal-aware LLaMA-Adapter (Zhang et al., 2023) achieves the best overall results, even outperforming the fully fine-tuned encoder-decoder model. We also empirically demonstrate that directly supervising the NLE generation process on the encoder latent space at the sentence level is viable, thus consolidating this approach as a first step towards the RL-free adversarial generation of NLEs when no (or few) ground-truth NLEs are available.

## Acknowledgments

This work is financed by National Funds through the FCT - Fundação para a Ciência e a Tecnologia, I.P. (Portuguese Foundation for Science and Technology) within the project CAGING, with reference 2022.10486.PTDC (DOI 10.54499/2022.10486.PTDC), and through the Ph.D. Grant 2020.07034.BD.

## Appendix A. Qualitative Examples

In Figure 3 we present examples of the NLEs generated by all models. These examples show that, although using only a small percentage of trainable parameters, PEFT approaches produce NLEs that are consistent with NLEs produced by fully fine-tuned models (RATCHET, DPT, and Base AE). Furthermore, they corroborate the known fact that NLG metrics are not good measures of the correctness of a given text: the Base AE model produces the same sentence for all instances, which can be explained by the high imbalance of MIMIC-NLE (evidence labels are predominantly 'Lung Opacity') (Kayser et al., 2022). For example, consider the last example where 'Atelectasis' is not present in the labels, but the NLE generated by the Base AE mentions that 'There is a new opacity in the right lung base which may represent atelectasis'. However, the majority of the NLG metrics (BERTScore, METEOR, BLEU-4, ROUGE, and SPICE) for the Base AE surpass those of the MSE loss trained model. On the other hand, both versions of the multi-modal LlaMA-Adapter present higher CLEV scores, indicating that they contain the correct clinical evidence, even though the syntax of the generated NLEs might deviate more from the ground-truth NLEs.

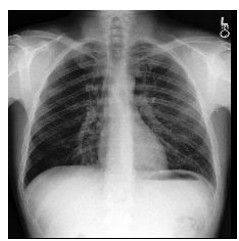

**Labels: Atelectasis** (U), Lung Opacity (P) and Pneumonia (U)

**GT:** There is new opacity at the right cardiophrenic angle, which may be atelectasis but could also represent pneumonia in the appropriate clinical setting.

**RATCHET:** Subtle opacity in the left lower lobe is concerning for pneumonia.

**DPT:** New right lower lobe opacity concerning for pneumonia.

**Base AE:** There is a new opacity in the right lung base which may represent atelectasis, but cannot exclude pneumonia or aspiration in the right clinical setting.

**LoRA:** New patchy opacities in the right upper lobe are concerning for pneumonia.

**Prompt:** Patchy opacity in the left lower lobe is likely due to atelectasis, but infection cannot be excluded in the appropriate clinical setting.

**Prefix:** There is a new focal opacity in the right upper lobe, concerning for pneumonia or aspiration in the appropriate clinical setting.

**LLaMA Uni:** Subtle opacity in the left lower lobe is likely atelectasis versus scarring in the appropriate clinical setting, a possible right middle lobe pneumonia cannot be excluded.

**LLaMA Mult:** Subtle opacity at the right lung base likely represents atelectasis or pneumonia.

**LLaMA Mult MSE:** There is patchy left basilar opacity likely reflecting atelectasis, but appear free air.

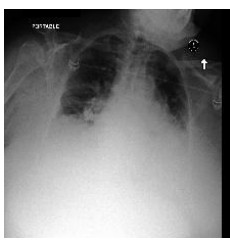

**Labels:** Atelectasis (U), **Edema** (U), Lung Opacity (P), Pleural Effusion (U) and Pneumonia (U)

**GT:** Retrocardiac opacity which may be due in part to pulmonary edema and atelectasis limits evaluation of the left lower lung, and superimposed pneumonia cannot be excluded.

**RATCHET:** There is prominence of the pulmonary vasculature consistent with mild pulmonary edema.

**DPT:** There is likely due to pulmonary edema.

**Base AE:** There is a new opacity in the right lung base which may represent atelectasis, but cannot exclude pneumonia or aspiration in the right clinical setting.

**LoRA:** There is mild interstitial prominence, which could reflect mild pulmonary edema, but not entirely excluded.

**Prompt:** There is mild interstitial prominence of the hila bilaterally, consistent with pulmonary edema.

**Prefix:** There is persistent bilateral perihilar opacities, likely representing asymmetric edema.

**LLaMA Uni:** There is persistent mild interstitial edema.

**LLaMA Mult:** There is increased interstitial markings in the left lower lobe, which may represent a component of pulmonary edema and atelectasis, less likely pneumonia.

**LLaMA Mult MSE:** There is hazy associated opacity in the left base likely representing atelectasis.

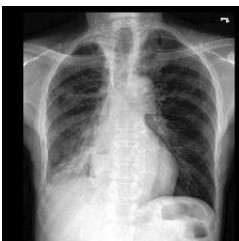

**Labels:** Lung Opacity (P) and **Pneumonia** (P)

**GT:** A new right lower lobe airspace opacity is likely due to aspiration pneumonia.

**RATCHET:** Right lower lobe consolidation is worrisome for pneumonia.

**DPT:** New right upper lobe opacity concerning for pneumonia.

**Base AE:** There is a new opacity in the right lung base which may represent atelectasis, but cannot exclude pneumonia or aspiration in the right clinical setting.

**LoRA:** There is persistent consolidation in the right upper lobe, which may represent aspiration or pneumonia in the appropriate clinical setting, worsening of the right lung base, and could be due to aspiration or pneumonia

**Prompt:** New opacification at the right lung base is compatible with pneumonia.

**Prefix:** There is a new focal opacity in the right upper lobe, concerning for pneumonia or aspiration.

**LLaMA Uni:** The opacity in the right upper lobe is concerning for pneumonia or aspiration, and less likely atelectasis.

**LLaMA Mult:** There is a new heterogeneous opacification in the right mid and lower lung zones concerning for pneumonia.

**LLaMA Mult MSE:** Given that the right basilar opacity is concerning for pneumonia, less likely atelectasis given the clinical setting.

Figure 3: Illustrative examples of NLEs generated by all tested models for correctly predicted images. For each image we provide the ground-truth labels, where: U - uncertain and P - positive. The labels highlighted in bold correspond to the disease being explained in the NLE.

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
