# OpenReview forum: "Parameter-Efficient Generation of Natural Language Explanations for Chest X-ray Classification"
_MIDL.io/2024/Conference — MIDL 2024 Oral_

### Official Review · Reviewer_R54G · 2024-02-27

**Confidence:** 4
**Preliminary Rating:** 4
**Final Rating:** 4

**Summary:**

This work extends upon previous work on NLE generation for multi-label chest X-ray diagnosis by replacing the traditional decoder-only NLE generator with an encoder-decoder architecture. The results show that using the multi-modal-aware LLaMA-Adapter achieves the best overall results

**Strengths:**

1. Based on the extension of previous work, the experimental results are relatively reliable.
2. The model architecture is introduced clearly.
3. The experiments include the ablation study of different models.

**Weaknesses:**

1. The model does not seem to perform better than previous methods on AUC and other metrics.
2. Whether this improvement has a medical motivation or is it the assembly of a model, a detailed explanation of the motivation is lacking.
3. Given some specific examples to explain your results would be better.

**Detailed Comments:**

1. Page 7 says: "We use a weight of 5 for the classification loss for all models, except for the model that uses MSE for the NLE loss, for which the weight is 2." How is this weight parameter set?
2. The model does not seem to perform better than previous methods on AUC and some other metrics. Please provide a detailed explanation.
3. Given some specific examples to explain your results would be better. Especially the comparison between your method and the previous method.

**Justification Of Final Rating:**

This paper extends previous work on NLE generation for multi-label chest X-ray diagnosis by replacing the traditional decoder-only NLE generator with an encoder-decoder architecture. This would be a contribution to the AI model explanation method. The experiments in this work are sufficient and the results presented are relatively good. Taken together, this work deserves to be accepted.

**Justification Of The Preliminary Rating:**

1. This work extends upon previous work on NLE generation for multi-label chest X-ray diagnosis by replacing the traditional decoder-only NLE generator with an encoder-decoder architecture, which seems a progress of NLE.
2. The model does not seem to perform better than previous methods on AUC and some other metrics, which may reduce trust in the validity of the model.
3. Lack of specific experimental examples to support experimental results.

**Questions To Address In The Rebuttal:**

1. Page 7 says: "We use a weight of 5 for the classification loss for all models, except for the model that uses MSE for the NLE loss, for which the weight is 2." How is this weight parameter set?
2. The model does not seem to perform better than previous methods on AUC and some other metrics. Please provide a detailed explanation.
3. Given some specific examples to explain your results would be better. Especially the comparison between your method and the previous method.

---

> ### Author Response · Authors · 2024-03-18
>
> We thank the reviewer for their insightful comments stressing that the proposed model architecture is clearly introduced and that an ablation study of different models is provided.
> &nbsp;
>
> &nbsp;
> ### Model performance
> Please refer to the general comment for a more in depth discussion concerning model performance, especially the AUC.
> &nbsp;
>
> &nbsp;
> ### Lack of motivation
> We clarified the motivation for replacing the decoder-only NLE generator by an encoder-decoder architecture and refer the reviewer to the general comment where this is further discussed.
> &nbsp;
>
> &nbsp;
> ### Classification loss weight
> All hyperparameters were chosen using the validation set of MIMIC NLE. As in Kayser et al., the best model corresponds to the one with the best overall results, i.e. the product between AUC, CLEV, and the average of BERTScore and METEOR. We clarified this in the updated manuscript.
> &nbsp;
>
> &nbsp;
> ### Qualitative examples
> We included some examples and further discussion of the NLEs generated by the different approaches in the supplementary material.

---

> > ### Comment · Reviewer_R54G · 2024-03-26
> >
> > Thank you for your response and clarifications, I am satisfied with the replies and have no further comments.

---

### Official Review · Reviewer_QYz9 · 2024-02-29

**Confidence:** 3
**Preliminary Rating:** 4
**Recommendation:** Poster
**Final Rating:** 4

**Summary:**

The authors propose a mostly exploratory style work, investigating various parameter efficient fine tuning methods for natural language explanations of chest X-ray classifications. From a technical novelty perspective, the authors build on one of the only NLE approaches so far tested for medical imaging and replace the decoder only style approach to an encoder-decoder style. The authors show this change allows them to only update a small fraction of parameters without needing RL to achieve strong results.

**Strengths:**

1. This work explores a fairly wide set of popular PEFT methods from the LLM (large language model) space. Figure 2 does a great job of showing these and their explanations while perhaps a bit on the short side (which makes sense given space limitations) are sufficient.

2. This is an exploratory work in a very interesting emerging area. With almost no comparable works, the authors do a good job over covering a fairly wide range of techniques. There is strong motivation for NLE's and with the massive recent advances in LLM's this work is very welcomed.

**Weaknesses:**

Given that metrics for NLE's are always very weak at best. It would be great to find a way to fit in some representative examples of NLE's generated by the various methods. I understand this is incredibly tough with the space limitations, but if accepted, I would strongly recommend including these in a supplementary information section (if allowed) or at least on an arXiv version of this paper.

Given a weak accept instead of strong because there isn't really any technical novelty in this work. That's fine, there are other types of papers than just novel techniques (for example exploration), but without any technical novelty, it's hard to support a very strong accept.

**Detailed Comments:**

N/A

**Justification Of Final Rating:**

Thanks to the authors for the comments. I stand by my original rating. I think this would make a great poster at MIDL, and maybe even meet the bar for an oral presentation due to the wide interest I expect the see from the community for this explorative work. I still disagree with the authors on how much technical novelty there is here, but that's of little consequence, the paper is a solid accept either way.

**Justification Of The Preliminary Rating:**

This is a very new space with few works in the area, and it is of incredible relevance and clinical importance. Exploratory works under these conditions are very welcomed. The only real shortcoming of this paper is there is no technical novelty introduced. I still find plenty of value to justify publication, but that is holding me back from giving a strong accept.

**Questions To Address In The Rebuttal:**

See weaknesses. If there is more technical novelty, laying this out in the contributions would be helpful. Just switching from a decoder only to an encoder-decoder (which already exists) isn't a technical novelty.

---

> ### Author Response · Authors · 2024-03-18
>
> We thank the reviewer for their insightful comments highlighting that our work is relevant to a “very interesting emerging area”, is “of incredible relevance and clinical importance” and “very welcomed”, and it covers a wide range PEFT techniques.
> &nbsp;
>
> ### Qualitative examples
> We included some examples and further discussion of the NLEs generated by the different approaches in the supplementary material.
> &nbsp;
>
> &nbsp;
>
>
> ### Novelty
> Although we recognise our work does not constitute a technical novelty in the sense that it does not propose a completely new architecture, we believe the novelty of this work lies in the fact that it is, to the best of our knowledge, the first to investigate PEFT for NLE generation in the medical domain. Furthermore, we also believe that this constitutes the first attempt at replacing the decoder-only architecture by an encoder-decoder framework for NLE generation and it paves the way for interesting future possibilities, mainly if one does not have ground-truth NLEs available during training. Please refer to the general comment where we dive deeper into these aspects. Throughout the updated manuscript (and in the contributions at the end of Section 1) we reformulated the parts regarding the motivation for the encoder-decoder architecture and the MSE loss experiment so that they are clearer.

---

### Official Review · Reviewer_8Pm1 · 2024-03-05

**Confidence:** 4
**Preliminary Rating:** 3
**Recommendation:** Poster
**Final Rating:** 4

**Summary:**

The authors develop a new model for natural langage explanations of chest x-ray classification using a a langage model conditioned on features from a vision model and the classification results. They train using parameter efficient methods and ablate the different methods and losses.

**Strengths:**

1. Good ablation of Parameter-efficient Fine-Tuning (PEFT):
- the number of PEFT methods is increasing steadily but no grounding and ablations have been made in the medical text-generation field. This is a valuable insight for future work as model size increases and compute osts become an issue.
2. Good experiments on which visual conditioning technique is the best:
- multi-modal langage generation is becoming more and more prominent and finding the most efficient way to condition text-generation to visual feature is still an open question.

**Weaknesses:**

1. The paper is a little bit difficult to follow:
- The MSE loss is introduced but no explanation about what it is applied on.
- The author talk about their method allowing to train the NLE model without reinforcement learning but no baseline use RL.
- The author introduce their method as using an auto-encoder both in the main text and in the Table 1 but not all methods are auto-encoder based according the Figure 2. The encoder is a BERT model and the decoder a transformer layer, but are the text tokens inputed in the BERT model or the decoder?
2. The reproduction of the baselines is significantly lower than in the original paper: it seems that both this paper and the original Kayser et al. paper (https://arxiv.org/pdf/2207.04343.pdf) use the same splits so the reproductions should match. This hinder the possibility to get conclusive results as DPT is said to be better than RATCHET according to this paper’s results but the reverse is true in the Kayser et al. paper.
3. The paper lacks better discussion on the results: Why « LLaMA-Adapter produces the best overall results » but is falling behind LoRA AE for all classical text metrics? It seems to be a trade-off between CLEV/BERTS and other metrics. Results are interesting as-is but the conclusion should be more nuanced and clearly explain why a metric is more important than the other (maybe with a cherry-picked example).

**Detailed Comments:**

I think that the experiments made by the author contains valuable insight for the community, however I think authors had issues reproducing the original DenseNet-121.
Kayser et al., authors of the DPT paper, say that they used an open implementation from TorchXRayVision, maybe switching with this version could help.  Reducing the size of figure 2 to make a 2-columns figure could help increase visibility and gain some space to increase the discussion.

The MSE loss is badly described and doesn’t add a lot of value to the paper. If I understand correctly, the author apply it between the output of the visually conditioned BERT encoder and on the output of the BERT encoder applied on the target NLE.
This is also possible to do so in the original Kayser et al. architecture by using the penultimate layer. If I make a mistake here, please add why in the main text.

I suggest to either remove it totally OR to introduce it as a way to boost results and to use both the Cross-Entropy loss and MSE loss combined, if it increases results like BERT score.

**Justification Of Final Rating:**

- Good ablation of Parameter-efficient Fine-Tuning (PEFT): the number of PEFT methods is increasing steadily but no grounding and ablations have been made in the medical text-generation field. This is a valuable insight for future work as model size increases and compute osts become an issue.
- Good experiments on which visual conditioning technique is the best: multi-modal langage generation is becoming more and more prominent and finding the most efficient way to condition text-generation to visual feature is still an open question.

My main concerns have been addressed and I think that the paper can add value to the NLE community.

**Justification Of The Preliminary Rating:**

The raw results are interesting and the medical field is lacking ablations on which classical multi-modal adapter works the best.
However the structure and clarity of the paper can be increased and would make me upgrade my rating.

**Questions To Address In The Rebuttal:**

1. Reformulate some of the passage about MSE, auto-encoder and reinforcement learning
2. Extend the discussion about the results

Optional:
3. I get that doing the experiments again using a better DenseNet is time consuming but baseline results should match or you should explain why they don't (DenseNet is trained with less epochs/less tricks etc.). If the paper is accepted and you manage to make results match, updating the table in the camera-ready version would be great.

---

> ### Author Response · Authors · 2024-03-18
>
> We thank the reviewer for the valuable comments and suggestions indicating that our work fills a gap in the medical imaging field by providing ablations on which multi-modal adaptation framework works best. With this rebuttal we aim to clarify our work, especially regarding the MSE loss, the discussion of the obtained results, and the reproduction of the chosen baselines.
> &nbsp;
>
> &nbsp;
>
> ### MSE Loss
> Throughout the paper we reformulated the passages regarding the motivation for the MSE loss experiment and how it is conducted (i.e., where the loss is applied on). Please refer to the general comment where we clarify the connection between the MSE Loss experiment, the choice regarding the replacement of the decoder-only model by the auto-encoder architecture, and the statement that no RL would be needed.
> &nbsp;
>
> &nbsp;
>
> ### Figure 2
> We clarified Figure 2, making it more explicit that our proposal uses an encoder-decoder architecture, setting it apart from RATCHET and DPT.
> &nbsp;
>
> &nbsp;
>
> ### Reproduction of results
> Please refer to the general comment for further discussion of this topic.
> &nbsp;
>
> &nbsp;
>
> ### Discussion of results and qualitative examples
> We included some examples of the NLEs generated by the different approaches in the supplementary material, together with an extended discussion (both in the main paper and in the appendix) regarding the relative importance of the NLG metrics and the CLEV score. Namely, even though some NLG metrics might be lower for some models, indicating that the syntax of their generated NLEs might be different from the one used in ground-truth NLEs, the clinical evidence remains correct.

---

> > ### Comment · Reviewer_8Pm1 · 2024-03-27
> >
> > Thank you for your satisfactory answers, I will improve my rating accordingly.

---

### Author Response · Authors · 2024-03-12

Dear Area Chairs,

In our rebuttal it would be very helpful if we could submit a pdf with some of the images requested by the reviewers.
Is it possible to do so? And if yes, how do you suggest we do that since we cannot directly upload the file via open review.

Thank you in advance.

---

> ### Comment · Area_Chair_1Axf · 2024-03-13
>
> Dear authors,
>
> You should be able to upload an updated version of your paper. Perhaps you can add an Appendix with the images?
>
> If you mean you want to show some images but not include them in the supplementary material, I suppose since the process is single-blind, you could put them on a public repository on Github, or some other platform which does not track the viewer (so do not use Google Docs for example).

---

> > ### Author Response · Authors · 2024-03-17
> >
> > Thank you for the clarification. We will add the necessary figures in the supplementary material of the updated version of the paper.

---

### Author Response · Authors · 2024-03-18
**Rebuttal - General Response**

We thank the reviewers for their valuable comments. As summarized by all reviewers, we propose replacing the traditional decoder-only architecture for generating Natural Language Explanations (NLEs) with an encoder-decoder trained with Parameter-Efficient Fine-Tuning (PEFT). The reviewers highlighted that our experimental methodology is well conducted, with sound ablation studies, the exploration of a wide range of PEFT methods, and that the topic is relevant for the medical imaging community.

This rebuttal addresses all reviewers’ concerns and the manuscript has been updated to include qualitative examples, further clarifications regarding the MSE loss, how the encoder-decoder architecture might avoid the need for Reinforcement Learning (RL), and a more extensive discussion of the results.
&nbsp;
&nbsp;

### Encoder-decoder and MSE Loss
Reviewer 8Pm1 asked for further clarification regarding the MSE loss, reviewer  QYz9 mentioned a lack of technical novelty (although still finding “plenty of value to justify publication”), and reviewer R54G states that the manuscript lacks a detailed explanation of the motivation. We believe these three concerns are connected and, thus, we answer them here. The MSE loss experiment is a preliminary test which paves the way and hints at the viability of a future architecture where no (or few) ground-truth NLEs are available during training, but no RL is needed. In this case (when teacher forcing is unfeasible), an adversarial approach can be used. However, if the generator is a decoder-only architecture (e.g. DPT), RL is needed to propagate approximate gradients from the discriminator to the generator, because the output of the decoder (which is also the input to the discriminator) is discrete. By using an encoder-decoder architecture, the adversarial learning happens on the continuous latent space of the encoder, so gradient propagation is ensured without needing RL, thus making the training process faster and more stable. Finally, decoding this latent space into text to obtain the NLE during testing is still possible thanks to the pre-trained decoder. Despite having ground-truth data available in this work, we wanted to test the viability of supervising the NLE generation directly in the output of the encoder, as this is where the adversarial learning would happen in the aforementioned RL-free adversarial NLE generation in low ground-truth availability regimes. We included this clarification throughout the updated manuscript, mainly in Section 2.3.
&nbsp;
&nbsp;

### Reproduction of results and AUC
We thoroughly revised our implementation and found that it was inconsistent with how Kayser et al. described the AUC computation. Additionally, through further experiments, we observed a big impact on the results of the weighting scheme between the classification and text losses. Given computational resource and time constraints, running an extensive hyperparameter search on the combination of these two losses was infeasible, so we opted for automatically learning the loss weights. We have updated the manuscript to include this weighting scheme and the updated results, which are now closer to the original work of Kayser et al.. Small differences might be due to the different loss weighting schemes and a different number of epochs. Moreover, NLG metrics are only computed on correctly classified instances, so small deviations on the classification side might have a big impact on the NLG metrics. Additionally, this makes it harder to compare with the work of Kayser et al.: even though two models might correctly classify the same number of instances, it might not be the same set of instances for the two models. The updated results show similar trends as before, with the multi-modal LlaMA-Adapter obtaining the best results among our models. The results of the MSE loss experiment improved, given the new automatic loss weighting scheme.

---

### Comment · Area_Chair_1Axf · 2024-03-18
**Invitation to reply to authors**

Dear reviewers,

The authors have prepared responses to your comments, which you should now be able to see in OpenReview. We encourage you to reply to their comments, and where necessary, adjust your rating. Please do so before the 27th of March.

---

### Meta-Review · Area_Chair_1Axf · 2024-04-01

**Recommendation:** Accept (Poster)
**Confidence:** 4

**Metareview:**

The paper addresses natural language explanations for chest x-ray classification by exploring how to tune such models more efficiently. The reviewers were fairly positive but raised some questions, in particular about replications of results from earlier work. This appears to be fixed in the revision, along with some other questions. Although the scores are not overwhelmingly positive I think I agree that this would make a good conference contribution.

---

### Decision · Program_Chairs · 2024-04-06

Accept (Oral)